# Nest-Site Features and Breeding Ecology of Chestnut-Vented Nuthatch *Sitta nagaensis* in Southwestern China

**DOI:** 10.3390/ani13122034

**Published:** 2023-06-19

**Authors:** Ruixin Mo, Yu Li, Qingmiao Yuan, Mingyun He, Xianyin Xu, Guangjian Chen, Wenwen Zhang, Yubao Duan

**Affiliations:** 1Key Laboratory for Conserving Wildlife with Small Populations in Yunnan, Southwest Forestry University, Kunming 650224, China; m15911476426@126.com (R.M.); yqm_24@163.com (Q.Y.); 2Department of Biodiversity Conservation, Southwest Forestry University, Kunming 650224, China; tencolumba@outlook.com; 3Administration of Zixi Mountain Provincial Nature Reserve, Chuxiong 675008, China; 18087521609@163.com (M.H.); tjt159694@163.com (X.X.); x124352023@163.com (G.C.); 4Research Center for Biodiversity Conservation and Biosafety/State Environmental Protection Key Laboratory on Biosafety, Nanjing Institute of Environmental Sciences, Ministry of Ecology and Environment of China, Nanjing 210042, China

**Keywords:** egg incubation, life history, nestling feeding, reproductive success, chestnut-vented nuthatch

## Abstract

**Simple Summary:**

Life history and its related theories are one of the most important topics in behavioral ecology, population ecology, and evolutionary biology. The reproductive ecology of birds is the cornerstone of avian life-history theory, and the success or failure of reproduction directly affects the survival and development of the population. This study investigated the reproductive ecology of chestnut-vented nuthatch (*Sitta nagaensis*) using artificial nest boxes to attract these secondary cavity-nesting birds. The study focused on a series of breeding activities and is the first breeding record of *S. nagaensis* in southwestern China, providing a basis for further reproductive ecology studies of this species.

**Abstract:**

The breeding ecology of birds is the cornerstone of bird life-history theory, and breeding success directly affects the survival and development of populations. We studied the breeding ecology of a secondary cavity-nesting bird, the chestnut-vented nuthatch *Sitta nagaensis*, in southwestern China from March to June in 2020, 2021, and 2022. In total, 16 nests in nest boxes and 19 nests in natural cavities were studied. The nesting habitat was mainly *Pinus yunnanensis* forest (68.4%), and the nest trees were mainly *P. yunnanensis* and pear *Pyrus* spp. Cavities made by woodpeckers and knot holes were used as nest sites, and the nuthatches plastered the hole entrance with mud. The nesting material was mainly pine bark. The clutch size was 3.47 ± 0.56 (range 2–4, *n* = 30), with an incubation period of 16.06 ± 0.91 days (range 15–19 days, *n* = 18). The nestling period was 20.88 ± 1.90 days (range 18–23 days, *n* = 23), and both parents fed the nestlings.

## 1. Introduction

Life history and its related theories are important topics in behavioral ecology, population ecology, and evolutionary biology [1,2]. A species-based life history is defined as the total set of processes by which a species grows, differentiates, reproduces, hibernates, and migrates. Among these, reproduction is considered the most important component of life history [3,4,5]. The success of reproduction directly affects the formation, extinction, and spread of species [6]. The basic reproductive ecology of birds, including the number of eggs, egg size, egg weight, breeding time, and survival rate and mortality of adults, is an important component of bird breeding research [5]. The rapid development of molecular technology and theory-based life-history approaches has led to increased research on the reproductive ecology of passerine birds, including life-history characteristics, nest-site selection, reproductive success rate, reproductive countermeasures, and offspring sex ratio, among others [7]. Zarybnicka et al. studied the relationship between nesting box occupation and habitat change in the boreal owl *Aegolius funereus* and found that the spawning period was positively correlated with the proportion of deciduous leaves around the nesting box [8]. Libois et al. found that using artificial nest boxes effectively increased the survival rate and reproductive success of the mew gull *Larus canus* [9]. Butler et al. reported that changes in the direction of the nest box had certain effects on nest box temperature and reproductive success rate [10]. The use of nesting boxes effectively increases the selection of nesting sites for hole-nesting birds and increases the breeding density, reducing to some extent the importance of the breeding habitat as the first element of breeding selection for birds [11].

The members of the Sittadiae, *Sitta villosa*, and *Sitta europaea* are important research subjects in China. Studies of the reproduction and feeding behavior of *S. europaea* showed that the nesting period is 10 or 11 days, the incubation period is 16 or 17 days, the brooding period is 18 or 19 days, the number of eggs laid is 8 or 9, and the chicks are mainly fed insects [12]. Additional research on the differences between the hatching input, number of nests, and growth and development of chicks of *S. europaea* showed that the breeding input of *S. europaea* could be adjusted according to its own body condition and the number of nest eggs, with 7 or 8 eggs being optimal [13,14].

The chestnut-vented nuthatch *Sitta nagaensis* is a medium-sized gray bird that inhabits coniferous and mixed-coniferous forests at altitudes of 1500–3000 m. It resembles *S. europaea* but has a pale-yellow underbody, and a gray throat, ear feathers, and breast, contrasting strongly with the dark brick red of its flanks. The undertail feathers are dark brown, with a distinct white scaly stripe on each side [15]. The species is mainly distributed in eastern and southeastern Tibet, southern and western Yunnan, northeastern Sichuan, and western and southwestern Guizhou. There are three subspecies of chestnut-vented nuthatch, namely the named subspecies (*S. nagaensis*), the southwest Asian species (*Sitta montiun*), and the Burmese subspecies (*Sitta grisiventris*).

In this study, we describe several aspects of the breeding biology of *S. nagaensis* breeding in natural cavities and nest boxes in southwestern China, such as nest site, nesting material, nest size, clutch size, egg-laying date, egg morphology, nestling development, and reproductive success. We compare the data from natural and nest-box nests to explore the impact of artificial nest boxes on the reproduction of this species. We discuss the effects of temperature on the length of the nestling period and competition for nest sites between this and other sympatric species. We also compare the life-history traits of *S. nagaensis* with its congeners to demonstrate the life-history strategy that contributes to its persistence in this region.

## 2. Materials and Methods

### 2.1. Study Area

The study site was located in Zixi Mountain Provincial Nature Reserve, Yunnan Province (101°22′49″–101°26′07″ E, 24°58′58″–25°04′01″ N, 1950–2502 m above sea level), with a total area of 160 km^2^ (Figure 1) [16]. Zixi mountain is the eastern branch of the Ailao mountains, with the highest middle ridge and two eastern and western slopes. It is located in the middle of the Yuanjiang River, Malone River, Jinsha River, and Longchuan River systems. The highest point is the main peak of Zixi mountain, with an altitude of 25,202 m, and the lowest point is at the outlet of the Maji River, with an altitude of 1950 m. Located in the monsoon belt of the northern subtropical plateau, the reserve has obvious seasonal changes, with abundant rainfall and high temperatures in summer, being relatively dry and with low temperatures in winter. The annual average precipitation is 900 mm, falling during the rainy season from May to October and the dry season from November to April of the following year. The frost period lasts 146–166 days, with a significant temperature difference between day and night. The average annual temperature is 12.1–14.9 °C [17]. Secondary coniferous and mixed-coniferous forests are the main vegetation types. The reserve has a forested area of 151.211 km^2^ and a nonforested area (including those not managed by forestry departments) of 24.92 km^2^. The main forest types in the reserve include *Castanopsis kawakami*, Yunnan pine, Huashan pine, and dry winter melon forests.

### 2.2. Field Work

The breeding ecology of *S. nagaensis* was studied from 2020 to 2022, using data collected from artificial nest boxes along with direct searches of natural nests. In total, 200 nest boxes were established between 2019 and 2020. Artificial nest boxes were mainly installed 3 or 4 months before the breeding season. The nest boxes were made of pine wood, 25 cm × 14 cm × 12 cm (height × length × width) in size, with a hole 5 cm in diameter. They were fixed to trees at 2.5–4 m above the ground. In total, 139 nest boxes (69.5%) were installed in coniferous forests, 44 (22%) in mixed forests, and 17 (8.5%) in plantations. The linear distance between each nest box was more than 50 m [18].

### 2.3. Data Collection

The nest boxes were checked every 2 days initially, and then checked daily when nesting materials were found in the boxes or when there was mud at the entrance. Searches were also made for tree cavities in the natural habitat (Figure 2c). Tree cavities were considered actively used if *S. nagaensis* visited frequently and were seen adding nesting materials. Miniature video cameras (HD99S-10, LooSafe, Dongguan, Guangzhou, China) were installed in four artificial nest-box nests in 2021 and were used to monitor the nests continuously from egg laying to when fledglings left the nest. The camera was positioned once the basic nest had been built and was fixed to the inside of the nest box in the center of the upper cover; the battery was fixed to the tree trunk under the nest box. Each camera was set to record video 24 h a day, with batteries replaced every 2 days. In total, 3504 h of videos were recorded.

Given that the length of the brooding period might be related to ambient temperature, weather conditions and average daily temperatures were also recorded throughout the study area. Temperature was measured using an electronic thermometer (WS-C, CKR, Penghe Electronics, Shanghai, China).

Habitat type was determined from a habitat type map showing the general layout of the nature reserve. For each nest tree, specimens of leaves and branches were collected and photographed. Nest materials were collected once the young had fledged. The species of each nest tree and the nesting materials used were confirmed against the Flora of China [19]. Egg color was determined by comparing standard colorimetric cards (PANTON formula guide coated and uncoated).

Active nests were visited every two days to determine the egg laying, hatching, and fledging dates. Adults were captured using a two-net set-up in front of the nest entrance or using a baffle to block the nest entrance, which then trapped them in the nest box. Each nestling was measured and ringed when they were 8–10 days old to ensure that individuals could be identified. In addition, egg size, nest size, nestling, and adult morphology were measured using vernier calipers accurate to 0.01 mm, without causing any damage to the birds. Nest height was measured using a measuring tape accurate to 0.01 m, and nest tree diameter at breast height (DBH) was measured using a diameter tape accurate to 0.01 cm. Egg mass, nestling mass, and adult mass were measured using an electronic scale accurate to 0.01 g. Egg mass was measured on the second day of the incubation period.

From the videos, records were made of the frequency and duration of daily incubation and nestling brooding by females from 06:00 h to 20:00 h, the frequency and timing of nest departures, the number of egg-turns per hour, the nestling feeding frequency per hour by parents, and the time that the females first exited and finally returned to the nest each day (individuals were identified by rings on their legs).

### 2.4. Data Analysis

Egg volume was calculated according to the formula of Hoyt [20]. The nestling period was divided into four equal periods [5]. Feeding data were taken for 1 day for each period, namely when nestlings were 1 day old, 8 days old, 15 days old, and 20 days old (the day before fledging) (Figure 3b,d,g,h). Body mass and tarsus measurements were selected to fit the growth and development curves of nestlings. Parameters related to reproductive outcomes included hatching success, fledging success, and reproductive success. Hatching success was defined as the proportion of eggs that hatched and was calculated based on all eggs; fledging success was defined as the proportion of nestlings that fledge and was calculated based on all successfully hatched nestlings; and reproductive success was defined as the proportion of nests that produced at least one fledgling [21].

Based on the video results, the duration from the moment the female returned to the nest to sit on eggs or nestlings to when it rose and left the nest was counted as the incubation or brooding time (the date and time were shown in the video data, accurate to the second). The frequency at which this behavior occurred in 1 h or 1 day was the hourly or daily incubation or brooding frequency, respectively. For egg-turning, any act of a female using her beak to pluck an egg or eggs was counted as egg-turning behavior; the number of times this behavior occurred in 1 h was the hourly egg-turning frequency.

Independent sample *t*-test and one-way ANOVA were used to detect differences, and Spearman’s correlation was used to test the correlation. Logistic regression was used to fit nestling growth curves. Data are presented as mean ± SD. Data analysis and graph production were performed using IBM SPSS 25.0 and R Studio 4.1.1. To not destroy the natural nests, detailed information on egg size, egg mass, nestling physical data, incubation behavior, nestling brooding, and feeding behavior were obtained from the nest-box nests.

## 3. Results

### 3.1. Nests and Nest-Site Features

In total, across 2020, 2021, and 2022, 16 nests in nest boxes and 19 nests in natural cavities were found in the study area (Table 1). Two nest-box nests in 2021 had only nesting materials but no eggs. One natural nest was occupied by a gray-headed woodpecker *Picus canus* before egg laying and, thus, was not included in the calculation of reproductive effectiveness.

The main nesting habitat was coniferous forests (accounting for 67.92% of the whole study area) in which the main tree species was *Pinus yunnanensis* (13/19 (68.4%) natural nests). The tree species used for natural cavity nests were *P. yunnanensis* (*n* = 7), *Pyrus* spp. (*n* = 5), *Alnus nepalensis* (*n* = 3), *Juglans regia* (*n* = 2), *Castanopsis orthacantha* (*n* = 1), and *Cunninghamia lanceolata* (*n* = 1). In their natural habitat, *S. nagaensis* nest in knot holes or cavities made by woodpeckers. The nesting materials in both the artificial and natural nests mainly comprised bark of *P. yunnanensis* and some fallen leaf fragments (Figure 3a). During the incubation period, the parents changed the structure of the bark stacks, arranging the bark around the eggs or nestlings from a flat surface to a shallow bowl shape (Figure 3b).

Mud was plastered to adjust the size of the entrance of the hole. The natural nest entrance was approximately oval in shape, with a long diameter of 26.45 ± 1.11 mm, a short diameter of 22.11 ± 1.02 mm (*n* = 18) (Figure 2b), and a nest depth of 226.29 ± 36.49 mm; they were sited 4.90 ± 2.99 m (*n* = 18) above the ground. *S. nagaensis* also used mud in the entrance of the nest boxes, making it smaller and approximately oval-shaped (Figure 2a). The long diameter of the entrance was 31.04 ± 3.03 mm, and the short diameter was 24.27 ± 1.25 mm (*n* = 14). There was a significant difference in the size of the nest entrance between the artificial and natural nests (*p* < 0.05, independent samples *t*-test). The modification of the entrance occurred mainly at the beginning of the breeding season, before egg laying. During the incubation period, parents also modified the entrance by partially peeling away the mud.

### 3.2. Egg Laying, Clutch Size, and Incubation Behavior

Data collected from all breeding nests showed that the earliest egg-laying date was from the end of March to early May. The egg-laying pattern was one egg per day. Clutch size was 3.47 ± 0.56 (range 2–4, *n* = 30) (Table 1), and egg color was white or light pink with purple-brown spots on the surface, mostly concentrated at the rounded end (Figure 3a). Egg size was 19.11 ± 0.62 mm × 14.15 ± 0.38 mm, egg mass was1.87 ± 0.09 g, and egg volume was 1936.94 ± 0.97 mm^3^ (*n* = 40). Only females incubated the eggs, and incubation began after the last egg had been laid. The incubation period lasted 16.06 ± 0.91 days (range 15–19 days, *n* = 18), and females stayed overnight in nests during the incubation period. The clutch size was not related to the egg-laying date (*p* > 0.05, Spearman’s correlation). There was no significant difference in the length of incubation period between the artificial and natural nests (*p* > 0.05, independent samples *t*-test) (Table 1). The egg-laying date in each study year was significantly earlier in natural nests than in nest-box nests (*p* < 0.05, independent samples *t*-test).

The first departure in the morning was between 06:30 h and 08:00 h, and the last departure was between 18:30 h and 19:40 h. Females covered eggs with nest materials when they left the nests. The number of times the female left the nest during the incubation period was 32.83 ± 8.38 per day, with a duration of 9.50 ± 8.12 min (range: 1–55 min) each time. The mean incubation per hour duration was 47.28 ± 3.94 min. The frequency of leaving the nest decreased, and the incubation time increased during the later stages of incubation (Figure 4), but there was no significant difference in frequency and duration of incubation and leaving the nest between the beginning (day 1) and the end of the incubation period (*p* > 0.05, one-way ANOVA). The females left the nest most frequently and for the longest duration between 13:00 h and 14:00 h (*p* < 0.01, one-way ANOVA) (Figure 5).

Hourly egg-turning frequency variation showed a similar pattern across the four nest-box nests videoed. The egg-turning frequency was lower during the early morning and evening and higher during the day, peaking at 10:00 h–11:00 h and 14:00 h–15:00 h (*p* < 0.05, one-way ANOVA), (Figure 6).

### 3.3. Nestling Brooding and Feeding Behavior

After the nestlings hatched, the female adults ingested the eggshells and remained sitting on the nestlings. As the nestlings grew, the duration and frequency of parental presence in the nests gradually decreased (Figure 4). There was a significant difference in hatching success between nest-boxes and natural nests (*p* < 0.05, one-way ANOVA) (Table 1). The daytime brooding behavior ceased once the nestlings were 15 days old, after which brooding behavior was not observed at any time of the day or night. Both parents participated in feeding the nestlings.

The nestlings produced fecal sacs and the parents swallowed these until the nestlings were 6 days old. After this point, the parents ejected the feces from the nests. Smaller nestlings showed escaping behavior by hiding under bark flakes; as they grew, the nestlings remained close to the back or corner of the nest box, away from the entrance. They only moved closer to the nest entrance when the adults were feeding them. Figure 7 shows that the hourly feeding frequency (no sex distinction) was higher between 08:00 h and 09:00 h, 10:00 h and 11:00 h, 13:00 h and 14:00 h, and 17:00 h and 18:00 h. The nestling stage lasted 20.88 ± 1.90 days (18–23 days, *n* = 23). There was no difference in nestling period length between natural and nest-box nests in the same year (*p* > 0.05, independent samples *t*-test). The length of the nestling period (all kinds of nests) varied significantly between years (*p* < 0.05, independent samples *t*-test) (Table 1).

The average temperature outside and the weather during the breeding season in each of the 3 years were calculated (Table 2). In 2022, the temperature was significantly lower than in other years, and the proportion of rainy and foggy days was higher (*p* < 0.05, one-way ANOVA).

### 3.4. Nestling Development

The growth curves of tarsus length and body mass of the fledglings were fitted by logistic regression (Figure 8). In total, 33 nestlings from nine nest boxes were measured daily, of which two nestlings died midway through the breeding season for unknown reasons, and their data were excluded from the model during calculations. The 1-day-old hatchlings were naked, with pink skin, closed eyes, and some fluffy down on the top of their heads and on their backs (Figure 3b). Feather buds appeared on the wings at 5 days old (Figure 3c), on the back and the tail at 8 days old (Figure 3d), and on the head at 10 days old (Figure 3e). From 13 days old (Figure 3f), the wing feathers started emerging from the capsules, and the tail feather at 15 days old (Figure 3g). When fledging, the plumage of the nestlings looked similar to that of the adults. The beak was light pink at hatching, then darkened gradually, and turned completely black by 14 days old. The feet and tarsi were also light pink at hatching, gradually turned yellow, and became tawny when fledged (Figure 3i), whereas the color of the feet and tarsus in adults was black. We compared four physical data between fledglings and adults (Table 3). There were significant differences in body length and bodyweight between fledglings and adults (*p* < 0.05, independent samples *t*-test). Fledglings were smaller in body length but heavier than adults. There was no significant difference in tarsus length between adults and fledglings (*p* > 0.05, independent samples *t*-test).

### 3.5. Breeding Success and Predation

Hatching success, fledging success, and reproductive success from natural and nest-box nests in each year were calculated separately (Table 1). Overall, the total reproductive success of 3 years of nest-box nests was 64.28% (9/14) and that of natural nests was 83.33% (15/18). The reproductive success of nest-box nests and natural nests did not vary significantly from year to year (*p* > 0.05, one-way ANOVA), but the total reproductive success of nest-box nests was significantly lower than that of natural nests (*p* < 0.05, independent samples *t*-test).

Among the three failed natural nests, two were because of the unexplained disappearance of the eggs, and the third was failure of the eggs to hatch. Among the five failed nest-box nests, one was attributed to damage to the nest box, three to the failure of egg hatching, and in the fifth nest, eggs were predated by a snake (later identified as *Elaphe carinata*, Figure 9).

## 4. Discussion

### 4.1. Nests and Nest-Site Features

In this study, it was found that *S. nagaensis* had a low utilization rate of artificial nest boxes, which might be because of the varying degree of adaptation of different bird species to artificial nest boxes. Previous work showed that *S*. *europaea* avoided nesting in nest boxes with suboptimally sized entrance holes and preferred natural nests over nest boxes [22]. We observed consistently fewer *S. nagaensis* using artificial nest boxes; in 2023, monitoring revealed nesting behavior in artificial nests without the presence of any eggs, which we consider tentative nesting behavior; this behavior might reflect an increase in the adaptability of *S. nagaensis* to artificial nest boxes, although further work is required to determine whether this is in fact the situation. The *S. nagaensis* that selected nest-box nests had later egg-laying dates compared with those using natural nests, which could be related to nest site competition. There is often fierce competition for high-quality nest sites [23]. Artificial nest boxes are relatively unfamiliar to birds and have unknown risks compared with natural tree holes and, thus, are usually not the optimal choice [24]. Thus, individuals that chose nest boxes might be at a disadvantage in nest site competition [25,26].

We observed that *Passer rutilans* cobred with *S. nagaensis* and competed for nest sites; their slightly larger body size allowed *P. rutilans* to easily win competitions for nest sites. When *S. nagaensis* inhabited a nest but had not yet laid eggs, *P. rutilans* tried to take over their nests (both natural and artificial nest-box nests). Therefore, *S. nagaensis* continued adding mud to further narrow the nest entrance, which prevented *P. rutilans* from entering the nest. However, *Picus canus* have beaks that are strong enough to peck through the mud at the entrance, enabling them to occupy the tree hole for their own breeding. In addition, the much larger body size of *P. canusr* compared with *S. nagaensis* enables the former to easily win the nest site competition. *S. europaea* also uses mud to coat the entrance to its nest hole, which is believed to help them defend nest encroachment by starlings [27,28]. We believe that this behavior can effectively avoid nest site competition among species and ensure the reproductive success of *S. nagaensis*, which is consistent with the hypothesis of Wesołowski and Rowiński [27] and Löhrl [29].

### 4.2. Breeding Behavior

Brooding patterns in *S. nagaensis* were unifemale, whereby males provided food to females during brooding but did not brood the eggs themselves. Incubation behavior is the result of a trade-off between the heat supplied to the eggs and the survival of parents, which affects the incubation rhythm and the foraging strategy chosen by the incubating parent [30,31]. For egg-turning frequency, the high frequency during two periods (10:00 h–11:00 h and 14:00 h–15:00 h) might result from the high ambient temperature. Turning the eggs allows them to be heated evenly and can effectively avoid adhesion of the embryo to the egg membrane, increasing hatching success [32]. Increasing the frequency of egg turning during periods of high ambient temperature could help *S. nagaensis* to heat eggs evenly and maintain optimal hatching temperatures. Moreover, the parents left the nest most frequently and for the longest duration between 13:00 h and 14:00 h (Figure 5). Given that the ambient temperature is the highest during this period (YL observation), adult birds frequently leave the nest for a long time, possibly because the optimal hatching temperatures can no longer be maintained by frequent egg turning; thus, it is necessary for the parent to leave the nest so that the temperature in the nest adjusts. The other option is that the eggs might cool less when the female leaves during the warmest periods of the day, which is a trade-off between reproduction and survival for females.

Food availability and ambient temperature can affect the growth of chicks in a given year [33]. Studies have shown that the brood period is relatively prolonged, so that the chicks can survive during low-temperature and high-precipitation conditions [7]. The brooding period of *S. nagaensis* was significantly different in different years, with that of 2021 being shorter than that of 2022. This might have something to do with the significant temperature differences between years. Temperatures were significantly lower in 2022 than in 2021, with a higher proportion of rainy and foggy days. The breeding behavior of *S. nagaensis* might be affected by the climate, which reduces the predation rate of parent birds and restrains the temperature maintenance of young birds; thus, *S. nagaensis* might adapt its breeding strategy accordingly. This is also illustrated by the high correlation between the brood length of *S. nagaensis* and the outdoor mean temperature and proportion of rain and fog during the breeding period. *S. nagaensis* mainly feeds on insects, and a relatively low external mean temperature might affect the active period of insects in the territory, reducing this food resource [21]. Given that the start of breeding of *S. nagaensis* did not differ from that of 2021, a decrease in food richness caused by low temperature could slow the growth of chicks. *S. nagaensis* has a relatively prolonged chick period, which helps to ensure the survival rate of chicks. In addition, less reproductive experience could also lead to a delay in the laying date of the first eggs. We also found nest loyalty in *S. nagaensis*, with breeding pairs using natural nests having a high probability of returning to the previous year’s nest in the second year; adult birds in natural nests were also likely to have more reproductive experience, whereas ring records indicated that breeding pairs that selected artificial nesting boxes were younger (1–2 years old) and less experienced, which might also be the reason for the later laying date of the first egg in artificial nests. However, because of a lack of data, further research is needed for verification.

For cave-nesters, it is risky to build a nest in a shallow hole, which is easier for predators to reach; deeper nests prevent predators from preying on nestlings and eggs, because they have to fit through the entrance hole to the nest. *S. nagaensis* nestlings showed avoidance behavior away from the nest entrance during the transition and independence stages. This behavior might occur for a similar reason to that seen in nestling *S*. *europaea*, namely to avoid predators. As the nestlings grew, the length and frequency of females warming them gradually decreased; thus, staying away from the nest entrance and being closer to each other might be beneficial for nestlings to maintain their body temperature in the nest, resulting in a higher survival rate.

Artificial nest boxes are relatively unfamiliar to birds and have unknown risks compared with natural tree holes and, thus, are usually not the optimal choice [24] and might have been defeated one or more times in nest-site competitions. This might result in a significant difference in their start breeding date compared with those occupying high-quality natural nests, with a delay in the date of laying the first egg. At the same time, less reproductive experience might also result in the first egg-laying date being later [34]; we found that five breeding pairs that chose artificial nest boxes were young couples (age determined by rings). The reproductive success of nest-box nests was less than that of natural nests, with the most important causative factors likely to be the instability of the nest box and the failure of egg hatching. Compared with natural nests, the walls of the nest boxes were thinner, the wooden structures were more susceptible to rain damage, the temperature inside the nest box might decrease more than in a natural nest, or the nest box might have been damaged (artificial damage), resulting in damaged eggs and/or failed hatching. However, because of the small sample size, no firm conclusions can be drawn and, thus, further work is required.

## 5. Conclusions

*S. nagaensis* preferred coniferous forest dominated by *P. yunnanensis* as its natural nesting habitat. The main nest-tree species was *P. yunnanensis*, the nest-hole type was pecking or knotholes, and the nesting material was mainly bark and fallen leaves. During the incubation period, the parents changed the structure of the bark stacks, arranging the bark around the eggs or nestlings from a flat surface to a shallow bowl shape, and mud was plastered to adjust the size of the entrance of the hole. Only female *S. nagaensis* incubated the eggs, whereas the males fed the females, leaving the nest several times for a short time.

*S. nagaensis* uses a biparental care strategy to rear its nestlings. As the nestlings grew, the time and frequency of parental brooding gradually decreased. The development of the chicks showed the following trends: the skin of the nestlings was naked on day 1, flight feathers begin to emerge at 13 days old, and tail feathers appeared from the feather tubes at 15 days old. The growth and development of the nestlings followed a logistic curve. When they fledged, the plumage of the fledglings was similar to that of the adults. There were significant differences in breeding period length and reproductive success rate between natural and artificial nests in different years, but no significant differences in reproductive success rate between different years.

## Figures and Tables

**Figure 1 animals-13-02034-f001:**
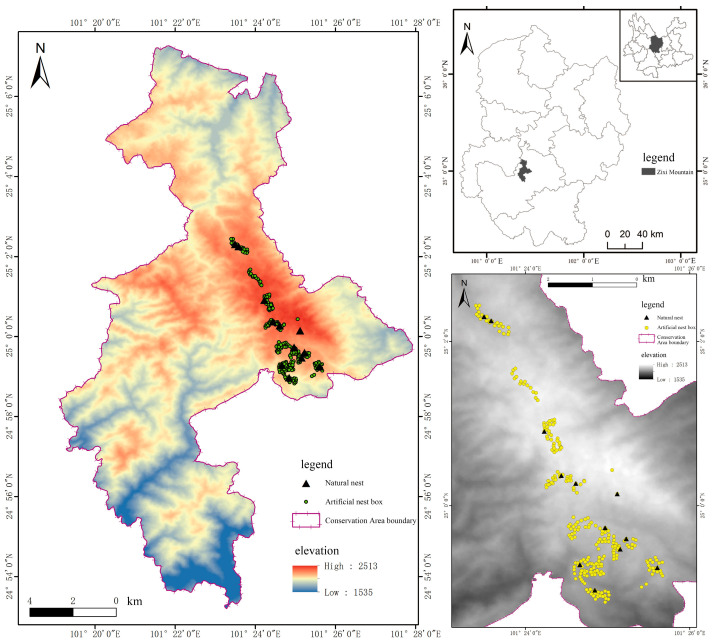
Overview of the study site.

**Figure 2 animals-13-02034-f002:**
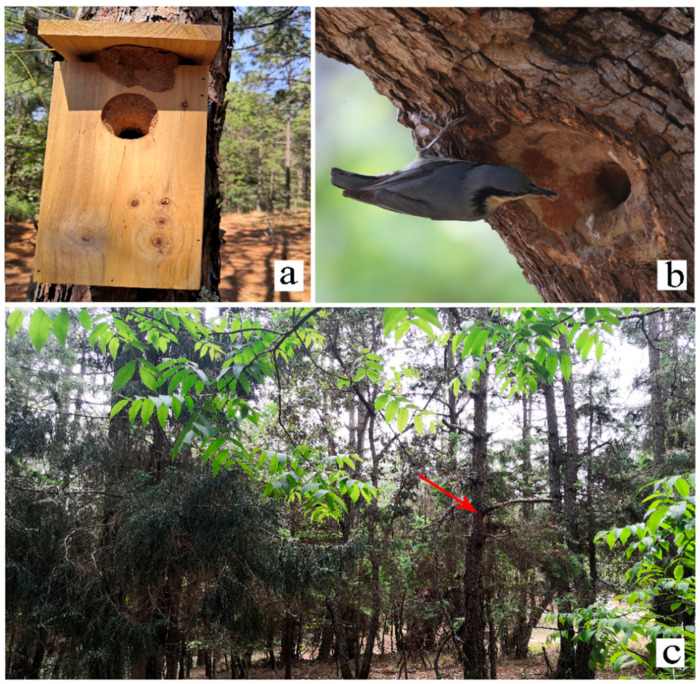
Nest of chestnut-vented nuthatch *Sitta nagaensis*: (**a**) nest box smeared with mud; (**b**) adult *S. nagaensis* and a natural nest in a pear *Pyrus* tree, smeared with mud; and (**c**) natural nesting environment of *S. nagaensis*.

**Figure 3 animals-13-02034-f003:**
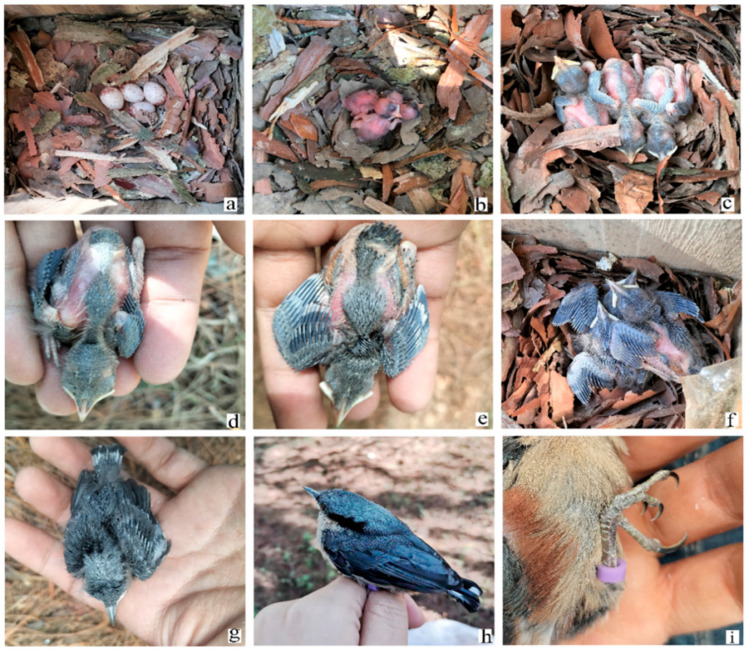
Nests, eggs, and nestlings of chestnut-vented nuthatch *Sitta nagaensis*: (**a**) nesting material and eggs in a nest box; (**b**) 1-day-old nestlings; (**c**) 5-day-old nestlings; (**d**) 8-day-old nestling; (**e**) 10-day-old nestling; (**f**) 13-day-old nestling; (**g**) 15-day-old nestling; (**h**) young the day before fledging (almost like adults); and (**i**) tarsus and claws of a nestling before fledging.

**Figure 4 animals-13-02034-f004:**
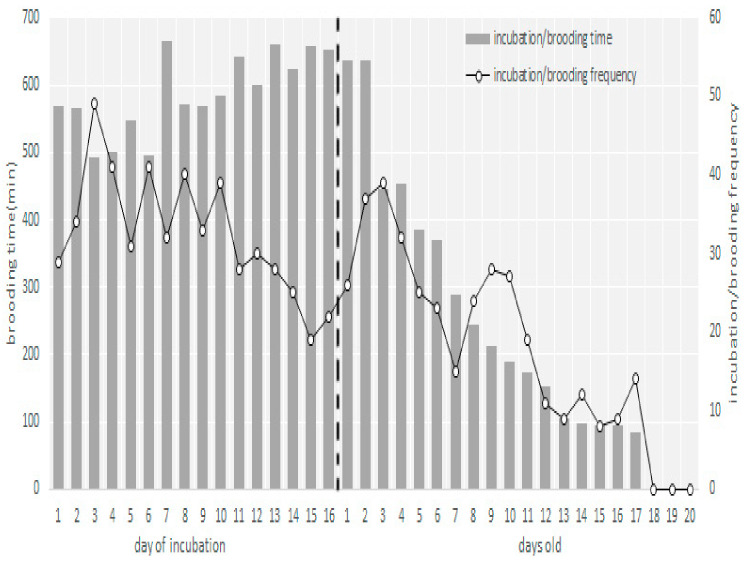
Daily frequencies and time of incubation and brooding: The incubation and brooding stages are separated by a dashed line. Incubation/brooding frequency refers to the frequency of the female parent warming eggs or nestlings.

**Figure 5 animals-13-02034-f005:**
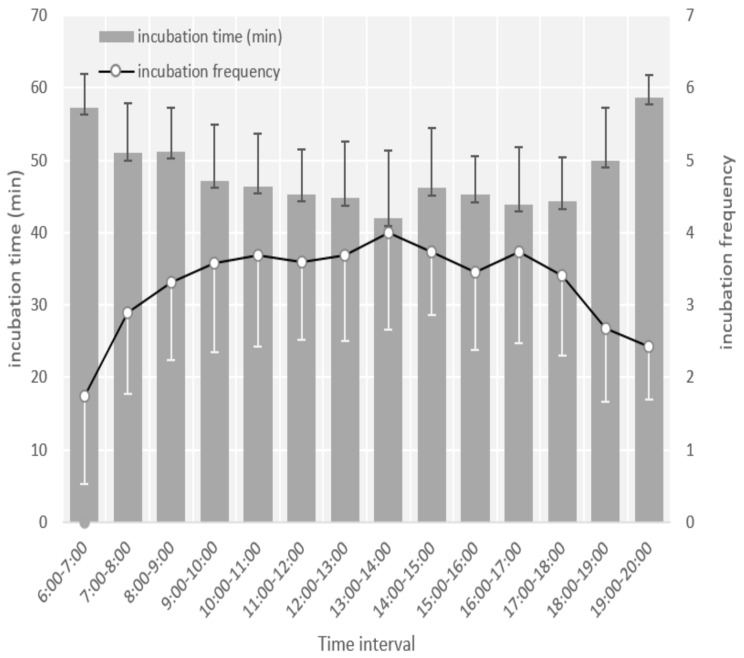
Hourly incubation frequencies and time of incubation during the daytime (06:00 h–20:00 h).

**Figure 6 animals-13-02034-f006:**
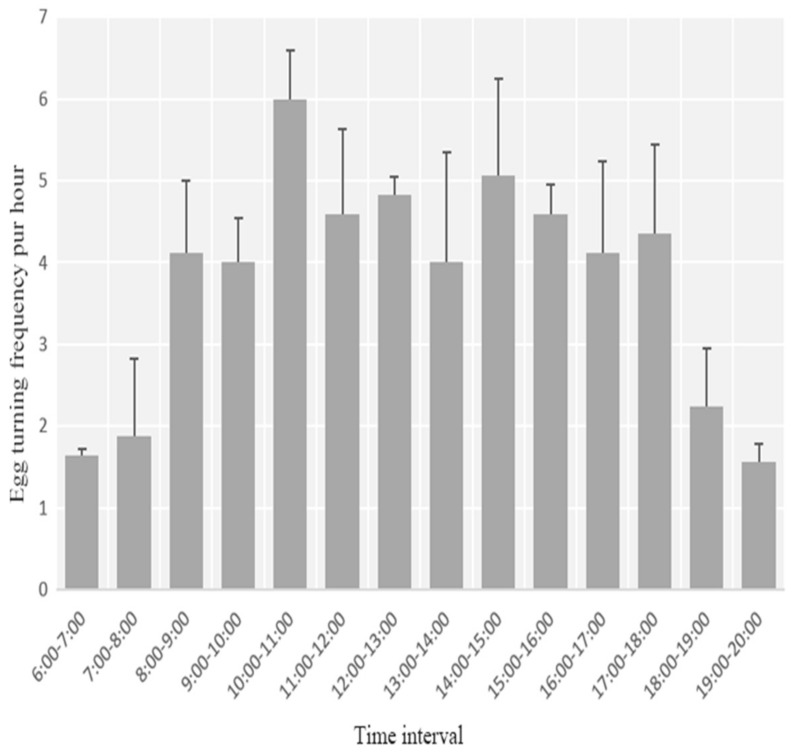
Egg-turning frequency per hour during the daytime (06:00 h–20:00 h).

**Figure 7 animals-13-02034-f007:**
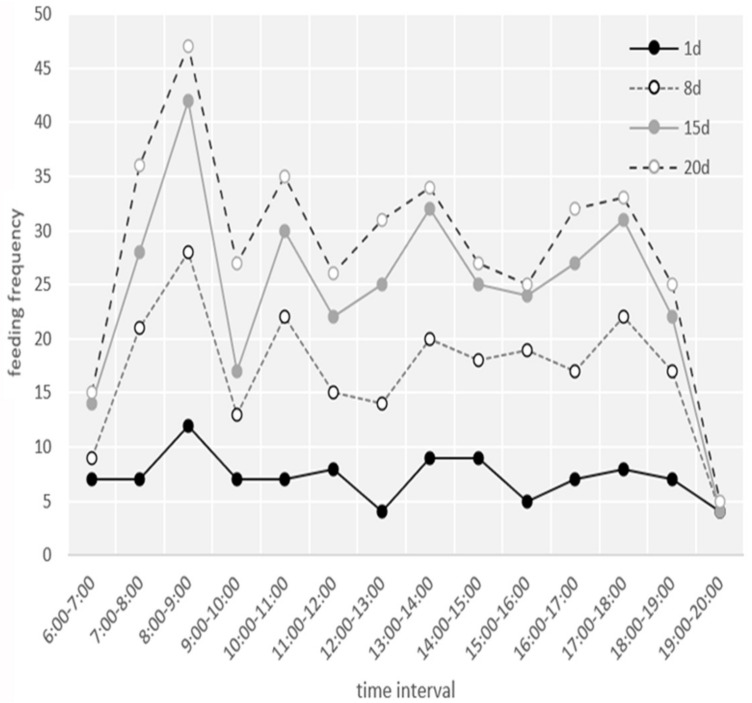
Feeding frequency per hour (06:00 h–20:00 h) during four stages of the brooding period: ‘20 d’ indicates the day the nestlings fledged.

**Figure 8 animals-13-02034-f008:**
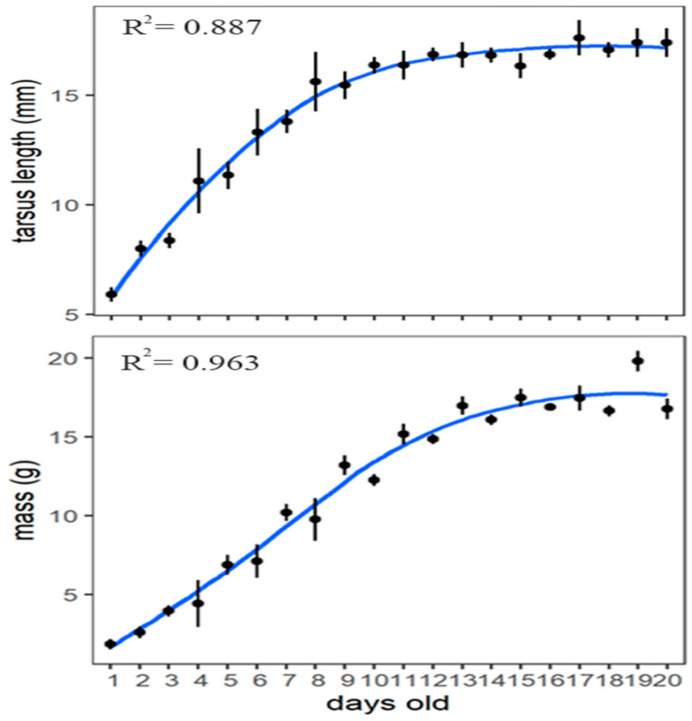
The mass and tarsus length development of *Sitta nagaensis* nestlings: there were different sample sizes of nestlings in the different periods, because some nestlings died after hatching.

**Figure 9 animals-13-02034-f009:**
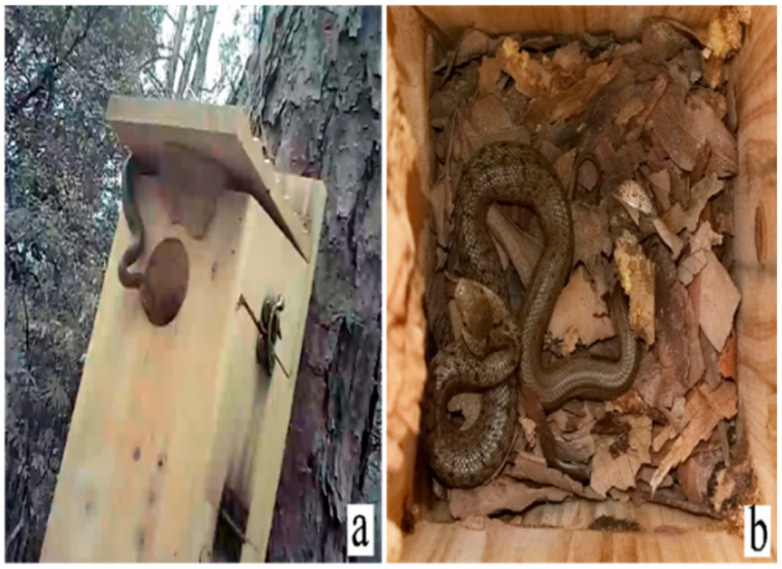
*Elaphe carinata*, a predator of the chestnut-vented nuthatch *Sitta nagaensis* in southwestern China: (**a**) the predator about to enter the nest; (**b**) predator eating eggs in nest.

**Table 1 animals-13-02034-t001:** Breeding characteristics of chestnut-vented nuthatch *S*. *nagaensis* in southwestern China.

Characteristic	2020		2021		2022	
	Natural nest	Nest-box nest	Natural nest	Nest-box nest	Natural nest	Nest-box nest
Number of nests	3	2	7	7	8	5
Clutch size (range)	4	3.5 ± 0.5	3.5 ± 0.76	3.43 ± 0.73	3.75 ± 0.43	3.6 ± 0.49
	4	3–4	2–4	3–4	3–4	3–4
Number of fledglings (all nests)	5	7	17	14	20	10
First egg laying date (earliest and latest)	27 March	5 April	26 March	15 April	25 March	11 April
	29 March	15 April	28 March	5 May	1 April	7 May
Hatching success	71.42%	87.5%	70.08%	58.33%	75%	52.63%
	5/7	7/8	17/24	14/24	20/26	10/19
Fledging success	100%	100%	76.47%	85.71%	100%	100%
	5/5	7/7	13/17	12/14	20/20	10/10
Reproductive success	100%	100%	85.71%	57.14%	87.5%	60%
	3/3	2/2	6/7	4/7	6/8	3/5

**Table 2 animals-13-02034-t002:** Average daily temperature and the proportion of rainy and foggy days during the nestling period.

Year	Average Daily Temperature	Proportion of Bad Weather *
2020	26.32 ± 2.74 °C range 12.8–23.6 °C, *n* = 55	32.72%
2021	25.99 ± 1.97 °C range 18.8–28.1 °C, *n* = 63	39.68%
2022	20.73 ± 4.25 °C range 12.8–23.6 °C, *n* = 71	45.07%

* Proportion of days with continuous rain or fog or both.

**Table 3 animals-13-02034-t003:** Average morphological characteristics of fledgling and adult *S. nagaensis*.

	Body Length (mm)	Bill Length (mm)	Tarsus Length (mm)	Mass (g)	Number
Adult	121.40 ± 4.94	16.16 ± 1.72	17.77 ± 1.40	14.79 ± 0.54	19
Fledgling	102.39 ± 4.44	14.44 ± 1.79	17.29 ± 0.52	17.51 ± 1.79	29

## Data Availability

All data were collected and analyzed by scientific methods in the field and can be provided if necessary.

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
