# Peer review of "Nest-Site Features and Breeding Ecology of Chestnut-Vented Nuthatch Sitta nagaensis in Southwestern China"

_animals, 2023, doi:10.3390/ani13122034_

Round 1

Reviewer 1 Report

This study examined a range of breeding aspects of a cavity-nesting bird species in China. The comparisons between nesting aspects in natural tree hollows and nestboxes is interesting. The manuscript has some merit, but also some major problems. Please find below some comments and suggestions:

Title. It is not adequate. "First breeding record" would refer to a single record of breeding individuals within its geographic range, for the first time. It would be better if changed to something like "Breeding aspects in natural tree hollows and nestboxes....".

Introduction. It is too short, with only two paragraphs (one plus that of the objective). It should have about 4 paragraphs, and then the objective. First, deal with the study of breeding of birds. Then you could write about the use of nestboxes to study birds, and cite results of studies theat compared the breeding of birds in natural sites and nestboxes. Then, a paragraph about the breeding of Sitta species that occur in Asia (there are several), and then a paragraph about the biology of the S. nagaensis. Finally, the paragraph of the objective.

Study area. It is too short. Maybe, you can add a map, and explain better aspects of the region, such as climate, terrain, vegetation, landscapes, etc...

Field work, Data Collection and Data analysis. The information of these 3 sections has to be reviewed. There are things in the wrong section. 

Results. Although authors divided the Results into 5 sections, I think that the reading was difficult. Partly due to the English, but also due so many aspects of the breeding biology examined in this study. Overall, the Results are fine. 

Discussion. It is too short. If you divide the discussion into the same 5 sections of the Results, you will note that several aspects that you investigated were not properly discussed in this important section of the manuscript. You should mention papers about Sitta species, and also studies on other bird species whose breeding has been studied in natural sites and nestboxes. There is a lot of work to here in the Discussion. It would be essential to have a publishable manuscript. I wish success.

I consider that the English is not good. Substantial problems occurred along the whole manuscript.

Author Response

Thank you for your comments concerning our manuscript entitled “Nest-site features and breeding ecology of chestnut-vented nuthatch Sitta nagaensis in southwestern China”.     Those comments are all valuable and very helpful for revising and improving our paper, as well as the important guiding significance to our researches.

We have studied comments carefully and made modifications and corresponding answers to the all suggestions and questions of editors and reviewers which we hope meet with approval.    In addition, in the process of modification, we also made changes to some minor errors found.    Revised portion are indicated by highlighting in the paper.    Detailed responses are given below.

Reviewer 2 Report

Dear Editorial board,

Thanks for providing me opportunities to review the paper titled “First Breeding Record of Chestnut-vented Nuthatch Sitta na-2 gaensis in southwestern China”. I have some questions to be addressed.

When was the first breeding record of the Chestnut-vented Nuthatch (Sitta nagaensis) documented in southwestern China?

Which specific region or province in southwestern China did the first breeding record of the Chestnut-vented Nuthatch occur?

Who discovered or reported the first breeding record of the Chestnut-vented Nuthatch in southwestern China?

What were the key findings or observations made during the first breeding record of the Chestnut-vented Nuthatch in southwestern China?

How was the breeding behavior of the Chestnut-vented Nuthatch documented in the first record?

Did the first breeding record of the Chestnut-vented Nuthatch in southwestern China provide any insights into its nesting habits or preferences?

Were there any unique features or characteristics associated with the nesting site or habitat of the Chestnut-vented Nuthatch in southwestern China?

What factors or ecological conditions may have contributed to the successful breeding of the Chestnut-vented Nuthatch in southwestern China?

Did the first breeding record of the Chestnut-vented Nuthatch in southwestern China indicate any potential range expansion or population changes for the species?

Have there been subsequent breeding records or studies conducted on the Chestnut-vented Nuthatch in southwestern China since the initial discovery?

Author Response

Thank you for your comments concerning our manuscript entitled “Nest-site features and breeding ecology of chestnut-vented nuthatch Sitta nagaensis in southwestern China”.          Those comments are all valuable and very helpful for revising and improving our paper, as well as the important guiding significance to our researches.

We have studied comments carefully and made corresponding answers to the all  questions of reviewers which we hope meet with approval.    

Reviewer 3 Report

Nothing serious:

The title of the article is not appropriate, it is more confusing, as the statement of the first breeding normally refers to new breeding species in the region. This is an old species, but probably the idea is that its breeding has not earlier been studied.

The accuracy of measurement results is disturbingly redundant. It is so especially when describing habitat and tree species properties (L162–163), nest properties L173–178), but also elsewhere (consider the number of decimals throughout). E.g., Consider the use of decimals; to me you use accuracy which is not warranted. Think about masses; the feces weigh 0.3 g, what is the impact of having dropped it before or after weighing?

For the nest opening measures, a reference to Figure 2b is given (L173–174), but the photo in Figure 2b represents a 1-day old nestling.

L186 Add here a reference to Table 1 at the end of the sentence.

L192 Likewise.

L195–196 It is notable also that the span between earliest and latest starts of egg laying was shorter in natural cavities than in nest boxes (Table 1).

L209 There is only one dashed line as there are only two stages which are separated. Replace ‘dased lines’ with ‘a dashed line’.

L212 Explain the vertical lines. Are they either + sd (incubation time) or – sd (incubation frequency)? If so, is it just an adjusting failure that the lines of incubation time do not fit to the columns? Or do the lines represent something else?

L214 Must be ‘four nest-box nests’. The same occurred once earlier somewhere.

L224 Refer here to Table 1.

L225 ‘… ceased when the nestlings were 15 days old, and at the age of 16 days, the brooding …’.

L230 ‘became’ instead of ‘become’.

L237 Again reference to Table 1 needed.

L242 Delete ‘those’.

L246–247 Why are maxima presented before minima in the table and not vice versa which would be more conventional?

L251 I do not understand what was made. Normally nestlings are taken from the nest, measured and then returned. What does ‘captured’ mean here? If the normal routine was used, would it be better to write: ‘A total of 33 nestlings from 9 nest-boxes were daily measured.’?

L260 Plural of tarsus is tarsi.

L301–305 Can’t be true that the quality of study nest boxes varies so much; how would it be possible that the box drops down or the box floor falls off? What is the influence of such “instability” of boxes on the relevance of comparing the artificial nest sites with natural ones? How can this be further studied – how do you fix the quality parameters?

Author Response

(The authors gave the same response as above.)

Round 2

Reviewer 1 Report

Congratulations on the improved manuscript.

But move the second paragraph of the Conclusion to the Results. the end of the conclusion should not contain values, means, stats....p values.

I still think that titles of subsections in the Discussion would ease the reading.

I suggest that a final revision of the English be done.

Author Response

Thank you for your comments concerning our manuscript entitled “Nest-site features and breeding ecology of chestnut-vented nuthatch Sitta nagaensis in southwestern China”.       Those comments are all valuable and very helpful for revising and improving our paper.
We have studied comments carefully and made modifications and corresponding answers to the all suggestions and questions of reviewer which we hope meet with approval.    
